# Adversarial Multi-Criteria Learning for Chinese Word Segmentation

## Abstract

Different linguistic perspectives causes many diverse segmentation criteria for Chinese word segmentation (CWS). Most existing methods focus on improve the performance for each single criterion. However, it is interesting to exploit these different criteria and mining their common underlying knowledge. In this paper, we propose adversarial multi-criteria learning for CWS by integrating shared knowledge from multiple heterogeneous segmentation criteria. Experiments on eight corpora with heterogeneous segmentation criteria show that the performance of each corpus obtains a significant improvement, compared to single-criterion learning.

## 1   Introduction

Chinese word segmentation (CWS) is a preliminary and important task for Chinese natural language processing (NLP). Currently, the state-of-the-art methods are based on statistical supervised learning algorithms (Xue, 2003; Zhao et al., 2006), and rely on a large-scale annotated corpus whose cost is extremely expensive. Although there have been great achievements in building CWS corpora, they are somewhat incompatible due to different segmentation criteria. As shown in Table 1, given a sentence "姚明进入总决赛 (YaoMing reaches the final)", the two commonly-used corpora, PKU's People's Daily (PKU) (Yu et al., 2001) and Penn Chinese Treebank (CTB) (Xia, 2000), use different segmentation criteria. In a sense, it is a waste of resources if we fail to fully exploit these corpora.

Recently, some efforts have been made to exploit heterogeneous annotation data for Chinese word segmentation or part-of-speech tagging

| Corpora | Yao | Ming | reaches | the final |
|---------|-----|------|---------|-----------|
| CTB | 姚明 | | 进入 | 总决赛 |
| PKU | 姚 | 明 | 进入 | 总 | 决赛 |

Table 1: Illustration of the different segmentation criteria.

(Jiang et al., 2009; Sun and Wan, 2012; Qiu et al., 2013; Li et al., 2015). These methods adopted stacking or multi-task architectures and showed that heterogeneous corpora can help each other. However, most of these model adopt the shallow linear classifier with discrete features, which makes it difficult to design the shared feature spaces, usually resulting in a complex model. Fortunately, recent deep neural models provide a convenient way to share information among multiple tasks (Collobert and Weston, 2008; Luong et al., 2015; Chen et al., 2016).

In this paper, we propose an adversarial multi-criteria learning for CWS by integrating shared knowledge from multiple segmentation criteria. Specifically, we regard each segmentation criterion as a single task and propose three different shared-private models under the framework of multi-task learning (Caruana, 1997; Ben-David and Schuller, 2003), where a shared layer is used to extract the criteria-invariant features, and a private layer is used to extract the criteria-specific features. Inspired by the success of adversarial strategy on domain adaption (Ajakan et al., 2014; Ganin et al., 2016; Bousmalis et al., 2016), we further utilize adversarial strategy to make sure the shared layer can extract the common underlying and criteria-invariant features, which are suitable for all the criteria. Finally, we exploit the eight segmentation criteria on the five simplified Chinese and three traditional Chinese corpora. Experiments show that our models are effective to improve the performance for CWS. We also observe

that traditional Chinese could benefit from incorporating knowledge from simplified Chinese.

The contributions of this paper could be summarized as follows.

- Multi-criteria learning is first introduced for CWS, in which we propose three shared-private models to integrate multiple segmentation criteria.
- An adversarial strategy is used to force the shared layer to learn criteria-invariant features, in which an new objective function is also proposed instead of the original cross-entropy loss.
- We conduct extensive experiments on eight CWS corpora with different segmentation criteria, which is by far the largest number of datasets used simultaneously.

## 2 General Neural Model for Chinese Word Segmentation

Chinese word segmentation task is usually regarded as a character based sequence labeling problem. Specifically, each character in a sentence is labeled as one of $\mathcal{L} = \{B, M, E, S\}$, indicating the begin, middle, end of a word, or a word with single character. There are lots of prevalent methods to solve sequence labeling problem such as maximum entropy Markov model (MEMM), conditional random fields (CRF), etc. Recently, neural networks are widely applied to Chinese word segmentation task for their ability to minimize the effort in feature engineering (Zheng et al., 2013; Pei et al., 2014; Chen et al., 2015; Ma and Hinrichs, 2015; Xu and Sun, 2016; Yao and Huang, 2016; Cai and Zhao, 2016; Zhang et al., 2016).

Specifically, given a sequence with $n$ characters $X = \{x_1, \ldots, x_n\}$, the aim of CWS task is to figure out the ground truth of labels $Y^* = \{y_1^*, \ldots, y_n^*\}$:

$$Y^* = \underset{Y \in \mathcal{L}^n}{\arg\max}\, p(Y|X), \qquad (1)$$

where $\mathcal{L} = \{B, M, E, S\}$.

The general architecture of neural CWS could be characterized by three components: (1) a character embedding layer; (2) feature layers consisting of several classical neural networks and (3) a tag inference layer. The role of feature layers is to extract features, which could be either convolution neural network or recurrent neural network. In this paper, we employ the state-of-the-art architecture

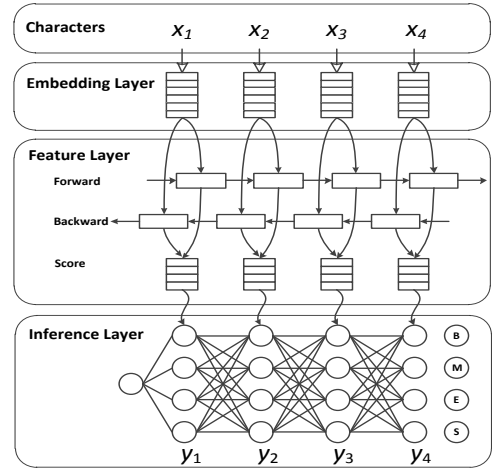

Figure 1: General neural architecture for Chinese word segmentation.

(Huang et al., 2015; Ma and Hovy, 2016) which adopts the bi-direction long short-term memory with CRF as tag inference layer. Figure 1 illustrates the general architecture of CWS.

### 2.1 Embedding layer

In neural models, the first step usually is to map discrete language symbols to distributed embedding vectors. Formally, we lookup embedding vector from embedding matrix for each character $x_i$ as $\mathbf{e}_{x_i} \in \mathbb{R}^{d_e}$, where $d_e$ is a hyper-parameter indicating the size of character embedding.

### 2.2 Feature layers

We adopt bi-directional long short-term memory (Bi-LSTM) as feature layers. While there are numerous LSTM variants, here we use the LSTM architecture used by (Jozefowicz et al., 2015), which is similar to the architecture of (Graves, 2013) but without peep-hole connections.

**LSTM** LSTM introduces gate mechanism and memory cell to maintain long dependency information and avoid gradient vanishing. Formally, LSTM, with input gate $\mathbf{i}$, output gate $\mathbf{o}$, forget gate $\mathbf{f}$ and memory cell $\mathbf{c}$, could be expressed as:

$$\begin{bmatrix} \mathbf{i}_i \\ \mathbf{o}_i \\ \mathbf{f}_i \\ \tilde{\mathbf{c}}_i \end{bmatrix} = \begin{bmatrix} \sigma \\ \sigma \\ \sigma \\ \phi \end{bmatrix} \left( \mathbf{W}_g^\mathsf{T} \begin{bmatrix} \mathbf{e}_{x_i} \\ \mathbf{h}_{i-1} \end{bmatrix} + \mathbf{b}_g \right), \quad (2)$$

$$\mathbf{c}_i = \mathbf{c}_{i-1} \odot \mathbf{f}_i + \tilde{\mathbf{c}}_i \odot \mathbf{i}_i, \qquad (3)$$

$$\mathbf{h}_i = \mathbf{o}_i \odot \phi(\mathbf{c}_i), \qquad (4)$$

where $\mathbf{W}_g \in \mathbb{R}^{(d_e+d_h) \times 4d_h}$ and $\mathbf{b}_g \in \mathbb{R}^{4d_h}$ are trainable parameters. $d_h$ is a hyper-parameter, in-

dicating the hidden state size. Function $\sigma(\cdot)$ and $\phi(\cdot)$ are sigmoid and tanh functions respectively.

**Bi-LSTM** In order to incorporate information from both sides of sequence, we use bi-directional LSTM (Bi-LSTM) with forward and backward directions. The update of each Bi-LSTM unit can be written precisely as follows:

$$\mathbf{h}_i = \overrightarrow{\mathbf{h}}_i \oplus \overleftarrow{\mathbf{h}}_i, \tag{5}$$

$$= \text{Bi-LSTM}(\mathbf{e}_{x_i}, \overrightarrow{\mathbf{h}}_{i-1}, \overleftarrow{\mathbf{h}}_{i+1}, \theta), \tag{6}$$

where $\overrightarrow{\mathbf{h}}_i$ and $\overleftarrow{\mathbf{h}}_i$ are the hidden states at position $i$ of the forward and backward LSTMs respectively; $\oplus$ is concatenation operation; $\theta$ denotes all parameters in Bi-LSTM model.

### 2.3 Inference Layer

After extracting features, we employ conditional random fields (CRF) (Lafferty et al., 2001) layer to inference tags. In CRF layer, $p(Y|X)$ in Eq (1) could be formalized as:

$$p(Y|X) = \frac{\Psi(Y|X)}{\sum_{Y' \in \mathcal{L}^n} \Psi(Y'|X)}. \tag{7}$$

Here, $\Psi(Y|X)$ is the potential function, and we only consider interactions between two successive labels (first order linear chain CRFs):

$$\Psi(Y|X) = \prod_{i=2}^{n} \psi(X, i, y_{i-1}, y_i), \tag{8}$$

$$\psi(\mathbf{x}, i, y', y) = \exp(s(X, i)_y + \mathbf{b}_{y'y}), \tag{9}$$

where $\mathbf{b}_{y'y} \in \mathbf{R}$ is trainable parameters respective to label pair $(y', y)$. Score function $s(X, i) \in \mathbb{R}^{|\mathcal{L}|}$ assigns score for each label on tagging the $i$-th character:

$$s(X, i) = \mathbf{W}_s^\top \mathbf{h}_i + \mathbf{b}_s, \tag{10}$$

where $\mathbf{h}_i$ is the hidden state of Bi-LSTM at position $i$; $\mathbf{W}_s \in \mathbb{R}^{d_h \times |\mathcal{L}|}$ and $\mathbf{b}_s \in \mathbb{R}^{|\mathcal{L}|}$ are trainable parameters.

## 3 Multi-Criteria Learning for Chinese Word Segmentation

Although neural models are widely used on CWS, most of them cannot deal with incompatible criteria with heterogonous segmentation criteria simultaneously.

Inspired by the success of multi-task learning (Caruana, 1997; Ben-David and Schuller, 2003),

we regard the heterogenous criteria as multiple "related" tasks, which could improve the performance of each other simultaneously with shared information.

Formally, assume that there are $M$ corpora with heterogeneous segmentation criteria. We refer $\mathcal{D}_m$ as corpus $m$ with $N_m$ samples:

$$\mathcal{D}_m = \{(X_i^{(m)}, Y_i^{(m)})\}_{i=1}^{N_m}, \tag{11}$$

where $X_i^m$ and $Y_i^m$ denote the $i$-th sentence and the corresponding label in corpus $m$.

To exploit the shared information between these different criteria, we propose three sharing models for CWS task as shown in Figure 2. The feature layers of these three models consist of a private (criterion-specific) layer and a shared (criterion-invariant) layer. The difference between three models is the information flow between the task layer and the shared layer. Besides, all of these three models also share the embedding layer.

### 3.1 Model-I: Parallel Shared-Private Model

In the feature layer of Model-I, we regard the private layer and shared layer as two parallel layers. For corpus $m$, the hidden states of shared layer and private layer are:

$$\mathbf{h}_i^{(s)} = \text{Bi-LSTM}(\mathbf{e}_{x_i}, \overrightarrow{\mathbf{h}}_{i-1}^{(s)}, \overleftarrow{\mathbf{h}}_{i+1}^{(s)}, \theta_s), \tag{12}$$

$$\mathbf{h}_i^{(m)} = \text{Bi-LSTM}(\mathbf{e}_{x_i}, \overrightarrow{\mathbf{h}}_{i-1}^{(m)}, \overleftarrow{\mathbf{h}}_{i+1}^{(m)}, \theta_m), \tag{13}$$

and the score function in the CRF layer is computed as:

$$s^{(m)}(X, i) = \mathbf{W}_s^{(m)\top} \begin{bmatrix} \mathbf{h}_i^{(s)} \\ \mathbf{h}_i^{(m)} \end{bmatrix} + \mathbf{b}_s^{(m)}, \tag{14}$$

where $\mathbf{W}_s^{(m)} \in \mathbb{R}^{2d_h \times |\mathcal{L}|}$ and $\mathbf{b}_s^{(m)} \in \mathbb{R}^{|\mathcal{L}|}$ are criterion-specific parameters for corpus $m$.

### 3.2 Model-II: Stacked Shared-Private Model

In the feature layer of Model-II, we arrange the shared layer and private layer in stacked manner. The private layer takes output of shared layer as input. For corpus $m$, the hidden states of shared layer and private layer are:

$$\mathbf{h}_i^{(s)} = \text{Bi-LSTM}(\mathbf{e}_{x_i}, \overrightarrow{\mathbf{h}}_{i-1}^{(s)}, \overleftarrow{\mathbf{h}}_{i+1}^{(s)}, \theta_s), \tag{15}$$

$$\mathbf{h}_i^{(m)} = \text{Bi-LSTM}(\begin{bmatrix} \mathbf{e}_{x_i} \\ \mathbf{h}_i^{(s)} \end{bmatrix}, \overrightarrow{\mathbf{h}}_{i-1}^{(m)}, \overleftarrow{\mathbf{h}}_{i+1}^{(m)}, \theta_m) \tag{16}$$

and the score function in the CRF layer is computed as:

$$s(X, i) = \mathbf{W}_s^{(m)\top} \mathbf{h}_i^{(m)} + \mathbf{b}_s^{(m)}, \tag{17}$$

where $\mathbf{W}_s^{(m)} \in \mathbb{R}^{2d_h \times |\mathcal{L}|}$ and $\mathbf{b}_s^{(m)} \in \mathbb{R}^{|\mathcal{L}|}$ are criterion-specific parameters for corpus $m$.

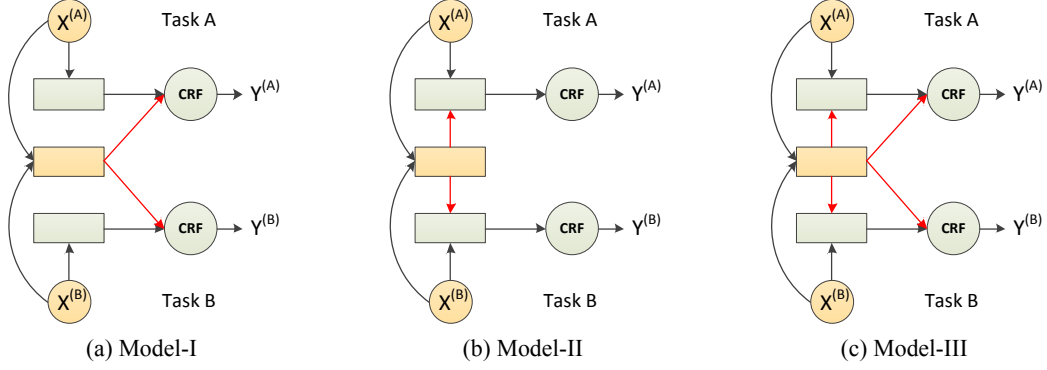

(a) Model-I (b) Model-II (c) Model-III

Figure 2: Three shared-private models for multi-criteria learning. The yellow blocks are the shared Bi-LSTM layer, while the gray block are the private Bi-LSTM layer. The yellow circles denote the shared embedding layer. The red information flow indicates the difference between three models.

### 3.3 Model-III: Skip-Layer Shared-Private Model

In the feature layer of Model-III, the shared layer and private layer are in stacked manner as Model-II. Additionally, we send the outputs of shared layer to CRF layer directly.

The Model III can be regarded as a combination of Model-I and Model-II. For corpus $m$, the hidden states of shared layer and private layer are the same with Eq (15) and (16), and the score function in CRF layer is computed as the same as Eq (14).

### 3.4 Objective function

The parameters of the network are trained to maximize the log conditional likelihood of true labels on all the corpora. The objective function $\mathcal{J}_{seg}$ can be computed as:

$$\mathcal{J}_{seg}(\Theta^m, \Theta^s) = \sum_{m=1}^{M} \sum_{i=1}^{N_m} \log p(Y_i^{(m)}|X_i^{(m)};\Theta^m,\Theta^s),$$

(18)

where $\Theta^m$ and $\Theta^s$ denote all the parameters in private and shared layers respectively.

### 4 Incorporating Adversarial Training for Shared Layer

Although the shared-private model separates the feature space into shared and private spaces, there is no guarantee that sharable features do not exist in private feature space, or vice versa. Inspired by the work on domain adaptation (Ajakan et al., 2014; Ganin et al., 2016; Bousmalis et al., 2016), we hope that the features extracted by shared layer is invariant across the heterogonous segmentation criteria. Therefore, we jointly optimize the shared

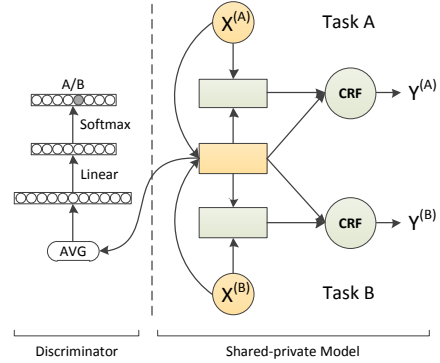

Figure 3: Architecture of Model-III with adversarial training strategy for shared layer. The discriminator firstly averages the hidden states of shared layer, then derives probability over all possible criteria by applying softmax operation after a linear transformation.

layer via adversarial training (Goodfellow et al., 2014).

Therefore, besides the task loss for CWS, we additionally introduce an adversarial loss to prevent criterion-specific feature from creeping into shared space as shown in Figure 3. We use a criterion discriminator which aims to recognize which criterion the sentence is annotated by using the shared features.

Specifically, given a sentence $X$ with length $n$, we refer to $\mathbf{h}_X^{(s)}$ as shared features for $X$ in one of the sharing models. Here, we compute $\mathbf{h}_X^{(s)}$ by simply averaging the hidden states of shared layer $\mathbf{h}_X^{(s)} = \frac{1}{n}\sum_i^n \mathbf{h}_{x_i}^{(s)}$. The criterion discriminator computes the probability $p(\cdot|X)$ over all criteria as:

$$p(\cdot|X;\Theta^d,\Theta^s) = \text{softmax}(\mathbf{W}_d^\top \mathbf{h}_X^{(s)} + \mathbf{b}_d), \quad (19)$$

where $\Theta^d$ indicates the parameters of criterion discriminator $\mathbf{W}_d \in \mathbb{R}^{d_h \times M}$ and $\mathbf{b}_d \in \mathbb{R}^M$; $\Theta^s$ denotes the parameters of shared layers.

## 4.1 Adversarial loss function

The criterion discriminator maximizes the cross entropy of predicted criterion distribution $p(\cdot|X)$ and true criterion.

$$\max_{\Theta^d} \mathcal{J}_{adv}^1(\Theta^d) = \sum_{m=1}^{M} \sum_{i=1}^{N_m} \log p(m|X_i^{(m)}; \Theta^d, \Theta^s). \quad (20)$$

An adversarial loss aims to produce shared features, such that a criterion discriminator cannot reliably predict the criterion by using these shared features. Therefore, we maximize the entropy of predicted criterion distribution when training shared parameters.

$$\max_{\Theta^s} \mathcal{J}_{adv}^2(\Theta^s) = \sum_{m=1}^{M} \sum_{i=1}^{N_m} H\left(p(m|X_i^{(m)}; \Theta^d, \Theta^s)\right), \quad (21)$$

where $H(p) = -\sum_i p_i \log p_i$ is an entropy of distribution $p$.

Unlike (Ganin et al., 2016), we use entropy term instead of negative cross-entropy.

## 5 Training

Finally, we combine the task and adversarial objective functions.

$$\mathcal{J}(\Theta; \mathcal{D}) = \mathcal{J}_{seg}(\Theta^m, \Theta^s) + \lambda \mathcal{J}_{adv}^1(\Theta^d) + \lambda \mathcal{J}_{adv}^2(\Theta^s), \quad (22)$$

where $\lambda$ is the weight that controls the interaction of the loss terms and $\mathcal{D}$ is the training corpora.

The training procedure is to optimize two discriminative classifiers alternately as shown in Algorithm 1. We use AdaGrad (Duchi et al., 2011) with minibatchs to maximize the objectives.

Notably, when using adversarial strategy, we firstly train 2400 epochs (each epoch only trains on eight batches from different corpora), then we only optimize $\mathcal{J}_{seg}(\Theta^m, \Theta^s)$ with $\Theta^s$ fixed until convergence (early stop strategy).

## 6 Experiments

### 6.1 Datasets

To evaluate our proposed architecture, we experiment on eight prevalent CWS datasets from SIGHAN2005 (Emerson, 2005) and SIGHAN2008 (Jin and Chen, 2008). Table 2 gives the details of the eight datasets. Among

---

**Algorithm 1** Adversarial multi-criteria learning for CWS task.

1: **for** $i = 1; i <= n\_epoch; i ++$ **do**
2: *# Train tag predictor for CWS*
3: **for** $m = 1; m <= M; m++$ **do**
4: *# Randomly pick data from corpus m*
5: $\mathcal{B} = \{X, Y\}_1^{b_m} \in \mathcal{D}^m$
6: $\Theta^s += \alpha \nabla_{\Theta^s} \mathcal{J}(\Theta; \mathcal{B})$
7: $\Theta^m += \alpha \nabla_{\Theta^m} \mathcal{J}(\Theta; \mathcal{B})$
8: **end for**
9: *# Train criterion discriminator*
10: **for** $m = 1; m <= M; m++$ **do**
11: $\mathcal{B} = \{X, Y\}_1^{b_m} \in \mathcal{D}^m$
12: $\Theta^d += \alpha \nabla_{\Theta^d} \mathcal{J}(\Theta; \mathcal{B})$
13: **end for**
14: **end for**

---

| | Datasets | | $N_w$ | $N_c$ | $|\mathcal{D}_w|$ | $|\mathcal{D}_c|$ | $N_s$ |
|---|---|---|---|---|---|---|---|
| Sighan05 | MSRA | Train | 2.4M | 4.1M | 88.1K | 5.2K | 86.9K |
| | | Test | 0.1M | 0.2M | 12.9K | 2.8K | 4.0K |
| | AS | Train | 5.4M | 8.4M | 141.3K | 6.1K | 709.0K |
| | | Test | 0.1M | 0.2M | 18.8K | 3.7K | 14.4K |
| Sighan08 | PKU | Train | 1.1M | 1.8M | 55.2K | 4.7K | 47.3K |
| | | Test | 0.2M | 0.3M | 17.6K | 3.4K | 6.4K |
| | CTB | Train | 0.6M | 1.1M | 42.2K | 4.2K | 23.4K |
| | | Test | 0.1M | 0.1M | 9.8K | 2.6K | 2.1K |
| | CKIP | Train | 0.7M | 1.1M | 48.1K | 4.7K | 94.2K |
| | | Test | 0.1M | 0.1M | 15.3K | 3.5K | 10.9K |
| | CITYU | Train | 1.1M | 1.8M | 43.6K | 4.4K | 36.2K |
| | | Test | 0.2M | 0.3M | 17.8K | 3.4K | 6.7K |
| | NCC | Train | 0.5M | 0.8M | 45.2K | 5.0K | 18.9K |
| | | Test | 0.1M | 0.2M | 17.5K | 3.6K | 3.6K |
| | SXU | Train | 0.5M | 0.9M | 32.5K | 4.2K | 17.1K |
| | | Test | 0.1M | 0.2M | 12.4K | 2.8K | 3.7K |

Table 2: Details of eight datasets. $N_w$ and $N_c$ indicate numbers of tokens and characters respectively. $\mathcal{D}_w$ and $\mathcal{D}_c$ are the dictionaries of distinguished words and characters respectively. $N_s$ indicates the number of sentences.

these datasets, AS, CITYU and CKIP are traditional Chinese, while the remains, MSRA, PKU, CTB, NCC and SXU, are simplified Chinese. We use 10% data of shuffled train set as development set for all datasets.

### 6.2 Experimental Configurations

Table 3 gives the details of the hyper-parameter configurations. Since the scale of each datasets varies, we use different training batch sizes for datasets. Specifically, we set batch sizes of AS and MSR datasets as 512 and 256 respectively, and 128 for remains. We employ dropout strategy on embedding layer, keeping 80% inputs (20% dropout rate).

| | |
|---|---|
| Character embedding size | $d_e = 50$ |
| Initial learning rate | $\alpha = 0.2$ |
| Loss weight coefficient | $\lambda = 0.05$ |
| LSTM dimensionality | $d_h = 100$ |
| Dropout rate on input layer | $p = 20\%$ |
| Batch size | 20 |

Table 3: Configurations of Hyper-parameters.

For initialization, we random all parameters following uniform distribution at $(-0.05, 0.05)$. We simply map traditional Chinese characters to simplified Chinese, and optimize on the same character embedding matrix across datasets, which is pre-trained on Chinese Wikipedia corpus, using word2vec toolkit (Mikolov et al., 2013). Following previous works (Chen et al., 2015; Pei et al., 2014), all experiments including baseline results are using pre-tarined character embedding with bi-gram feature.

### 6.3 Overall Results

Table 4 shows the experiment results of the proposed models on test sets of eight CWS datasets, which has three blocks.

(1) In the first block, we can see that the performance of Bi-LSTM cannot be improved by merely increasing the depth of networks.

(2) In the second block, our proposed three models based on multi-criteria learning boost performance. Model-I gains 0.75% improvement on averaging F-measure score compared with Bi-LSTM result (94.14%). Only the performance on MSRA drops slightly. Compared to the baseline results (Bi-LSTM and stacked Bi-LSTM), the proposed models boost the performance with the help of exploiting information across these heterogeneous segmentation criteria. Although various criteria have different segmentation granularities, there are still some underlying information shared. For instance, MSRA and CTB treat family name and last name as one token "宁泽涛 (NingZeTao)", whereas some other datasets, like PKU, regard them as two tokens, "宁 (Ning)" and "泽涛 (Ze-Tao)". The partial boundaries (before "宁 (Ning)" or after "涛 (Tao)") can be shared.

(3) In the third block, we introduce adversarial training. By introducing adversarial training, the performances are further boosted, and Model-I is slightly better than Model-II and Model-III. The adversarial training tries to make shared layer keep

criteria-invariant features. For instance, as shown in Table 4, when we use shared information, the performance on MSRA drops (worse than baseline result). The reason may be that the shared parameters bias to other segmentation criteria and introduce noisy features into shared parameters. When we additionally incorporate the adversarial strategy, we observe that the performance on MSRA is improved and outperforms the baseline results. We could also observe the improvements on other datasets. However, the boost from the adversarial strategy is not significant. The main reason might be that the proposed three sharing models implicitly attempt to keep invariant features by shared parameters and learn discrepancies by the task layer.

### 6.4 Traditional & Simplified Chinese

Traditional Chinese and simplified Chinese are two similar languages with slightly difference on character forms and usages on grammar. We investigate that if datasets in traditional Chinese and simplified Chinese could help each other. Table 5 gives the results of Model-I on 3 traditional Chinese datasets under the help of 5 simplified Chinese datasets. Specifically, we firstly train the model on simplified Chinese datasets, then we train traditional Chinese datasets independently with shared parameters fixed.

As we can see, the average performance is boosted by 0.41% on F-measure score (from 93.78% to 94.19%), which indicates that shared features learned from simplified Chinese segmentation criteria can help to improve performance on traditional Chinese. Like MSRA, as AS dataset is relatively large (train set of 5.4M tokens), the features learned by shared parameters might bias to other datasets and thus hurt performance on such large dataset AS.

### 6.5 Speed

To further explore the convergence speed, we plot the results on development sets through epochs. Figure 4 shows the learning curve of Model-I without incorporating adversarial strategy. As shown in Figure 4, the proposed model makes progress gradually on all datasets. After about 1000 epochs, the performance becomes stable and convergent.

We also test the decoding speed, and our models process 441.38 sentences per second averagely. As the proposed models and the baseline models (Bi-LSTM and stacked Bi-LSTM) are nearly in the same complexity, all models are nearly the

| Models | | MSRA | AS | PKU | CTB | CKIP | CITYU | NCC | SXU | Avg. |
|---|---|---|---|---|---|---|---|---|---|---|
| Bi-LSTM | P | 95.70 | 93.64 | 93.67 | 95.19 | 92.44 | 94.00 | 91.86 | 95.11 | 93.95 |
| | R | 95.99 | 94.77 | 92.93 | 95.42 | 93.69 | 94.15 | 92.47 | 95.23 | 94.33 |
| | F | **95.84** | 94.20 | 93.30 | **95.30** | **93.06** | **94.07** | 92.17 | 95.17 | **94.14** |
| | OOV | 66.28 | 70.07 | 66.09 | 76.47 | 72.12 | 65.79 | 59.11 | 71.27 | 68.40 |
| Stacked Bi-LSTM | P | 95.69 | 93.89 | 94.10 | 95.20 | 92.40 | 94.13 | 91.81 | 94.99 | 94.03 |
| | R | 95.81 | 94.54 | 92.66 | 95.40 | 93.39 | 93.99 | 92.62 | 95.37 | 94.22 |
| | F | 95.75 | **94.22** | **93.37** | **95.30** | 92.89 | 94.06 | **92.21** | **95.18** | 94.12 |
| | OOV | 65.55 | 71.50 | 67.92 | 75.44 | 70.50 | 66.35 | 57.39 | 69.69 | 68.04 |
| Multi-Criteria Learning | | | | | | | | | | |
| Model-I | P | 95.67 | 94.44 | 94.93 | 95.95 | 93.99 | 95.10 | 92.54 | 96.07 | 94.84 |
| | R | 95.82 | 95.09 | 93.73 | 96.00 | 94.52 | 95.60 | 92.69 | 96.08 | 94.94 |
| | F | 95.74 | 94.76 | **94.33** | 95.97 | **94.26** | 95.35 | **92.61** | 96.07 | **94.89** |
| | OOV | 69.89 | 74.13 | 72.96 | 81.12 | 77.58 | 80.00 | 64.14 | 77.05 | 74.61 |
| Model-II | P | 95.74 | 94.60 | 94.82 | 95.90 | 93.51 | 95.30 | 92.26 | 96.17 | 94.79 |
| | R | 95.74 | 95.20 | 93.76 | 95.94 | 94.56 | 95.50 | 92.84 | 95.95 | 94.94 |
| | F | 95.74 | **94.90** | 94.28 | 95.92 | 94.03 | 95.40 | 92.55 | 96.06 | 94.86 |
| | OOV | 69.67 | 74.87 | 72.28 | 79.94 | 76.67 | 81.05 | 61.51 | 77.96 | 74.24 |
| Model-III | P | 95.76 | 93.99 | 94.95 | 95.85 | 93.50 | 95.56 | 92.17 | 96.10 | 94.74 |
| | R | 95.89 | 95.07 | 93.48 | 96.11 | 94.58 | 95.62 | 92.96 | 96.13 | 94.98 |
| | F | **95.82** | 94.53 | 94.21 | **95.98** | 94.04 | **95.59** | 92.57 | **96.12** | 94.86 |
| | OOV | 70.72 | 72.59 | 73.12 | 81.21 | 76.56 | 82.14 | 60.83 | 77.56 | 74.34 |
| Adversarial Multi-Criteria Learning | | | | | | | | | | |
| Model-I+ADV | P | 95.95 | 94.17 | 94.86 | 96.02 | 93.82 | 95.39 | 92.46 | 96.07 | 94.84 |
| | R | 96.14 | 95.11 | 93.78 | 96.33 | 94.70 | 95.70 | 93.19 | 96.01 | 95.12 |
| | F | **96.04** | 94.64 | **94.32** | **96.18** | **94.26** | **95.55** | **92.83** | 96.04 | **94.98** |
| | OOV | 71.60 | 73.50 | 72.67 | 82.48 | 77.59 | 81.40 | 63.31 | 77.10 | 74.96 |
| Model-II+ADV | P | 96.02 | 94.52 | 94.65 | 96.09 | 93.80 | 95.37 | 92.42 | 95.85 | 94.84 |
| | R | 95.86 | 94.98 | 93.61 | 95.90 | 94.69 | 95.63 | 93.20 | 96.07 | 94.99 |
| | F | 95.94 | **94.75** | 94.13 | 96.00 | 94.24 | 95.50 | 92.81 | 95.96 | 94.92 |
| | OOV | 72.76 | 75.37 | 73.13 | 82.19 | 77.71 | 81.05 | 62.16 | 76.88 | 75.16 |
| Model-III+ADV | P | 95.92 | 94.25 | 94.68 | 95.86 | 93.67 | 95.24 | 92.47 | 96.24 | 94.79 |
| | R | 95.83 | 95.11 | 93.82 | 96.10 | 94.48 | 95.60 | 92.73 | 96.04 | 94.96 |
| | F | 95.87 | 94.68 | 94.25 | 95.98 | 94.07 | 95.42 | 92.60 | **96.14** | 94.88 |
| | OOV | 70.86 | 72.89 | 72.20 | 81.65 | 76.13 | 80.71 | 63.22 | 77.88 | 74.44 |

Table 4: Results of proposed models on test sets of eight CWS datasets. There are three blocks. The first block consists of two baseline models: Bi-LSTM and stacked Bi-LSTM. The second block consists of our proposed three models without adversarial training. The third block consists of our proposed three models with adversarial training.

| | AS | CKIP | CITYU | Avg. |
|---|---|---|---|---|
| Base line | **94.20** | 93.06 | 94.07 | 93.78 |
| This work | 94.12 | **93.24** | 95.20 | **94.19** |

Table 5: Performance on 3 traditional Chinese datasets with shared parameters fixed shared. The shared parameters are trained on 5 simplified Chinese datasets. Here, we conduct Model-I without incorporating adversarial training strategy.

same efficient. However, the time consumption of training process varies from model to model. For the models without adversarial training, it costs about 10 hours for training (the same for stacked Bi-LSTM to train eight datasets), whereas it takes about 16 hours for the models with adversarial training. All the experiments are conducted on the hardware with Intel(R) Xeon(R) CPU E5-2643 v3 @ 3.40GHz and NVIDIA GeForce GTX TITAN

X.

### 6.6 Error Analysis

We further investigate the benefits of the proposed models by comparing the error distributions between the single-criterion learning (baseline model Bi-LSTM) and multi-criteria learning (Model-I and Model-I with adversarial training) as shown in Figure 5. According to the results, we could observe that a large proportion of points lie above diagonal lines in Figure 5a and Figure 5b, which implies that performance benefit from integrating knowledge and complementary information from other corpora. As shown in Table 4, on the test set of CITYU, the performance of Model-I and its adversarial version (Model-I+ADV) boost from 92.17% to 95.59% and 95.42% respectively.

In addition, we observe that adversarial strategy is effective to prevent criterion specific features

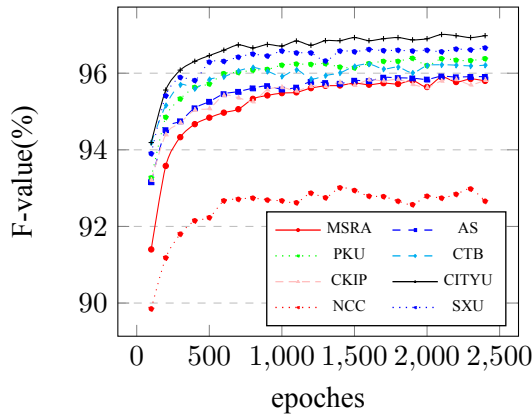

Figure 4: Convergence speed of Model-I without adversarial training on development sets of eight datasets.

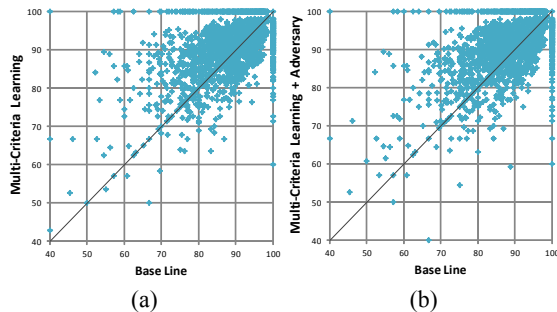

(a) (b)

Figure 5: F-measure scores on test set of CITYU dataset. Each point denotes a sentence, with the (x, y) values of each point denoting the F-measure scores of the two models, respectively. (a) is comparison between Bi-LSTM and Model-I. (b) is comparison between Bi-LSTM and Model-I with adversarial training.

from creeping into shared space. For instance, the segmentation granularity of personal name is often different according to heterogenous criteria. With the help of adversarial strategy, our models could correct a large proportion of mistakes on personal name. Table 6 lists the examples from 2333-th and 89-th sentences in test sets of PKU and MSRA datasets respectively.

## 7 Related Works

There are many works on exploiting heterogeneous annotation data to improve various NLP tasks. Jiang et al. (2009) proposed a stacking-based model which could train a model for one specific desired annotation criterion by utilizing knowledge from corpora with other heterogeneous annotations. Sun and Wan (2012) proposed a

| Models | PKU-2333 | | MSRA-89 |
|---|---|---|---|
| Golds | Lu 卢 | Wu Xuan 武铉 | Mu Ling Ying 穆玲英 |
| Base Line | 卢武铉 | | 穆 玲英 |
| Model-I | 卢武铉 | | 穆 玲英 |
| Modell-I+ADV | 卢 | 武铉 | 穆玲英 |

Table 6: Segmentation cases of personal name.

structure-based stacking model to reduce the approximation error, which makes use of structured features such as sub-words. These models are unidirectional aid and also suffer from error propagation problem.

Qiu et al. (2013) used multi-tasks learning framework to improve the performance of POS tagging on two heterogeneous datasets. Li et al. (2015) proposed a coupled sequence labeling model which could directly learn and infer two heterogeneous annotations. Chao et al. (2015) also utilize multiple corpora using coupled sequence labeling model. These methods adopt the shallow classifiers, therefore suffering from the problem of defining shared features.

Our proposed models use deep neural networks, which can easily share information with hidden shared layers. Chen et al. (2016) also adopted neural network models for exploiting heterogeneous annotations based on neural multi-view model, which can be regarded as a simplified version of our proposed models by removing private hidden layers.

Unlike the above models, we design three sharing-private architectures and keep shared layer to extract criterion-invariance features by introducing adversarial training. Moreover, we fully exploit eight corpora with heterogeneous segmentation criteria to model the underlying shared information.

## 8 Conclusions & Future Works

In this paper, we propose adversarial multi-criteria learning for CWS by fully exploiting the underlying shared knowledge across multiple heterogeneous criteria. Experiments show that our proposed three shared-private models are effective to extract the shared information, and achieve significant improvements over the single-criterion methods.

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
