# Peer review of "Adversarial Multi-Criteria Learning for Chinese Word Segmentation"

_ACL 2017 — decision unknown_

[Official Review · Reviewer 1 · rating 4 · confidence 3]
soundness 5 · originality 5 · clarity 3 · impact 4 · substance 5 · appropriateness 5 · meaningful comparison 4 · presentation format Poster

- Strengths:

The authors use established neural network methods (adversarial networks --
Goodfellow et al, NIPS-2014) to take advantage of 8 different Chinese work
breaking test sets, with 8 different notions of what counts as a word in
Chinese.

This paper could have implications for many NLP tasks where we have slightly
different notions of what counts as correct.  We have been thinking of that
problem in terms of adaptation, but it is possible that Goodfellow et al is a
more useful way of thinking about this problem.

- Weaknesses:

We need a name for the problem mentioned above.  How about: the elusive gold
standard.  I prefer that term to multi-criteria.

The motivation seems to be unnecessarily narrow.  The elusive gold standard
comes up in all sorts of applications, not just Chinese Word Segmentation.

The motivation makes unnecessary assumptions about how much the reader knows
about Chinese.              When you don't know much about something, you think it is
easier than it is.  Many non-Chinese readers (like this reviewer) think that
Chinese is simpler than it is.              It is easy to assume that Chinese Word
Segmentation is about as easy as tokenizing English text into strings delimited
by white space.  But my guess is that IAA (inter-annotator agreement) is pretty
low in Chinese.  The point you are trying to make in Table 1 is that there is
considerable room for disagreement among native speakers of Chinese.

I think it would help if you could point out that there are many NLP tasks
where there is considerable room for disagreement.  Some tasks like machine
translation, information retrieval and web search have so much room for
disagreement that the metrics for those tasks have been designed to allow for
multiple correct answers.  For other tasks, like part of speech tagging, we
tend to sweep the elusive gold standard problem under a rug, and hope it will
just go away.  But in fact, progress on tagging has stalled because we don't
know how to distinguish differences of opinions from errors.  When two
annotators return two different answers, it is a difference of opinion.  But
when a machine returns a different answer, the machine is almost always wrong.

This reader got stuck on the term: adversary.  I think the NIPS paper used that
because it was modeling noise under "murphy's law."  It is often wise to assume
the worst.

But I don't think it is helpful to think of differences of opinion as an
adversarial game like chess.  In chess, it makes sense to think that your
opponent is out to get you, but I'm not sure that's the most helpful way to
think about differences of opinion.

I think it would clarify what you are doing to say that you are applying an
established method from NIPS (that uses the term "adversarial") to deal with
the elusive gold standard problem.  And then point out that the elusive gold
standard problem is a very common problem.  You will study it in the context of
a particular problem in Chinese, but the problem is much more general than
that.

- General Discussion:

I found much of the paper unnecessarily hard going.  I'm not up on Chinese or
the latest in NIPS, which doesn't help.  But even so, there are some small
issues with English, and some larger problems with exposition.

Consider Table 4.  Line 525 makes an assertion about the first block and depth
of networks.  Specifically, which lines in Table 4 support that assertion.

I assume that P and R refer to precision and recall, but where is that
explained.  I assume that F is the standard F measure, and OOV is
out-of-vocabulary, but again, I shouldn't have to assume such things.

There are many numbers in Table 4.  What counts as significance?  Which numbers
are even comparable?  Can we compare numbers across cols?  Is performance on
one collection comparable to performance on another?  Line 560 suggests that
the adversarial method is not significant.  What should I take away from Table
4?  Line 794 claims that you have a significant solution to what I call the
elusive gold standard problem.              But which numbers in Table 4 justify that
claim?

Small quibbles about English:

works --> work (in many places).  Work is a  mass noun, not a count noun
(unlike "conclusion").              One can say one conclusion, two conclusions, but
more/less/some work (not one work, two works).

line 493: each dataset, not each datasets

line 485: Three datasets use traditional Chinese (AS, CITY, CKIP) and the other
five use simplified Chinese.

line 509: random --> randomize

[Official Review · Reviewer 2 · rating 4 · confidence 4]
soundness 5 · originality 5 · clarity 4 · impact 4 · substance 5 · appropriateness 5 · meaningful comparison 4 · presentation format Oral Presentation

The paper proposes a method to train models for Chinese word segmentation (CWS)
on datasets having multiple segmentation criteria.

- Strengths:
1. Multi-criteria learning is interesting and promising.
2. The proposed model is also interesting and achieves a large improvement from
baselines.

- Weaknesses:
1. The proposed method is not compared with other CWS models. The baseline
model (Bi-LSTM) is proposed in [1] and [2]. However, these model is proposed
not for CWS but for POS tagging and NE tagging. The description "In this paper,
we employ the state-of-the-art architecture ..." (in Section 2) is misleading.
2. The purpose of experiments in Section 6.4 is unclear. In Sec. 6.4, the
purpose is that investigating "datasets in traditional Chinese and simplified
Chinese could help each other." However, in the experimental setting, the model
is separately trained on simplified Chinese and traditional Chinese, and the
shared parameters are fixed after training on simplified Chinese. What is
expected to fixed shared parameters?

- General Discussion:
The paper should be more interesting if there are more detailed discussion
about the datasets that adversarial multi-criteria learning does not boost the
performance.

[1] Zhiheng Huang, Wei Xu, and Kai Yu. 2015. Bidirectional lstm-crf models for
sequence tagging. arXiv preprint arXiv:1508.01991.
[2] Xuezhe Ma and Eduard Hovy. 2016. End-to-end sequence labeling via
bi-directional lstm-cnns-crf. arXiv preprint arXiv:1603.01354 .